# SARS-CoV-2 co-detection with influenza and human respiratory syncytial virus in Ethiopia: Findings from the severe acute respiratory illness (SARI) and influenza-like illness (ILI) sentinel surveillance, January 01, 2021, to June 30, 2022

Wolde Shure[1], Adamu Tayachew[1], Tsegaye Berkessa[1]*, Gizaw Teka[1], Mengistu Biru[1], Ayele Gebeyehu[1], Adane Woldeab[1,2], Musse Tadesse[1], Melaku Gonta[1], Admikew Agune[1], Aster Hailemariam[1], Bizuwork Haile[1], Beza Addis[1], Muluken Moges[1], Leuel Lisanwork[2], Lehageru Gizachew[2], Eyasu Tigabu[2], Zelalem Mekuria[3], Getnet Yimer[4], Nebiyu Dereje[5], Jemal Aliy[5], Sileshi Lulseged[5], Zenebe Melaku[5], Ebba Abate[2], Wondwossen Gebreyes[3,6], Mesfin Wossen[1], Aschalew Abayneh[1]

1 Ethiopian Public Health Institute, Addis Ababa, Ethiopia, 2 The Ohio State University Global One Health initiative (GOHi), Addis Ababa, Ethiopia, 3 The Ohio State University Global One Health initiative (GOHi), Columbus, OH, Unites States of America, 4 Perelman School of Medicine, University of Pennsylvania, Philadelphia, PA, Unites States of America, 5 ICAP at Columbia University, Addis Ababa, Ethiopia, 6 Department of Veterinary Preventive Medicine, Infectious Diseases, The Ohio State University, Columbus, OH, Unites States of America

* tsegayebtola@gmail.com

## Abstract

SARS-CoV-2 co-infection with the influenza virus or human respiratory syncytial virus (RSV) may complicate its progress and clinical outcomes. However, data on the co-detection of SARS-CoV-2 with other respiratory viruses are limited in Ethiopia and other parts of Africa to inform evidence-based response and decision-making. We analyzed 4,989 patients' data captured from the national severe acute respiratory illness (SARI) and influenza-like illness (ILI) sentinel surveillance sites over 18 months period from January 01, 2021, to June 30, 2022. Laboratory specimens were collected from the patients and tested for viral respiratory pathogens by real-time, reverse transcription polymerase chain reaction (RT-PCR) at the national influenza center. The median age of the patients was 14 years (IQR: 1–35 years), with a slight preponderance of them being at the age of 15 to less than 50 years. SARS-CoV-2 was detected among 459 (9.2%, 95% CI: 8.4–10.0) patients, and 64 (1.3%, 95% CI: 1.0–1.6) of SARS-CoV-2 were co-detected either with Influenza virus (54.7%) or RSV (32.8%) and 12.5% were detected with both of the viruses. A substantial proportion (54.7%) of SARS-CoV-2 co-detection with other respiratory viruses was identified among patients in the age group from 15 to less than 50 years. The multivariable analysis found that the odds of SARS-CoV-2 co-detection was higher among individuals with the age category of 20 to 39 years as compared to those less than 20 years old (AOR: 1.98, 95% CI:1.15–3.42) while the odds of SARS-CoV-2 co-detection was lower among cases from

**Data Availability Statement:** All relevant data are within the paper and its Supporting Information files.

**Funding:** The authors received no specific funding for this work.

**Competing interests:** The authors have declared that no competing interests exist.

other regions of the country as compared to those from Addis Ababa (AOR:0.16 95% CI:0.07–0.34). Although the SARS-CoV-2 co-detection with other respiratory viral pathogens was minimal, the findings of this study underscore that it is critical to continuously monitor the co-infections to reduce transmission and improve patient outcomes, particularly among the youth and patients with ILI.

## Introduction

Viral respiratory tract infections are a major cause of illness and mortality around the world. Most of the cases and deaths occur among young children living in resource-constrained settings like Ethiopia [1]. However, the diversity of respiratory infections imposes a great challenge to controlling the problem. On the other hand, the group of viruses that can cause respiratory tract infections are highly dependent on seasonality and the change in climatic conditions. Hence, their prevention and control are challenging due to the high transmissibility and capacity to evolve and cause epidemics in sporadic regions and sometimes crossing boundaries and becoming pandemics as well [2, 3].

Previous studies have reported variable rates of co-infections between SARS-CoV-2 and other respiratory viruses ranging from 0 to 20% [4–10]. In Northern California, a study done by Kim et al. found that 20.7% of COVID-19 patients were co-infected with at least one other respiratory pathogen. Among co-infecting agents, respiratory viruses are the most prevalent. The most frequent co-infecting agents are respiratory viruses [7]. In Ecuador, co-infections with other respiratory viruses were detected in 12% of SARSCoV-2-positive patients. The most prevalent co-infection was with influenza virus at 4.4%, followed by respiratory syncytial virus with 3.1% [11].

According to a meta-analysis conducted by Lansbury et al. Influenza virus, respiratory syncytial virus (RSV), and adenovirus are common viral co-infections observed in individuals positive for SARS-CoV-2 [12]. A systematic review by Maltezou et al. revealed that the median percentage of SARS-CoV-2 and influenza co-infections was 4.9%, while the mean percentage was estimated to be 16.3% in patients with SARS-CoV-2 infection, ranging from 0.04% to 58.3% [13].

Co-detection of SARS-CoV-2 with other respiratory viruses depends on the dynamic of infection of each pathogen which adds to the challenge of diagnosing COVID-19 [14]. Since SARS-CoV-2 and other viral respiratory infections share the same laboratories, sentinel sites, and reporting platforms, the World Health Organization (WHO) promoted the identification of SARS-CoV-2. This was primarily accomplished through the use of the Global Influenza Surveillance and Response System (GISRS) [15]. In Ethiopia, influenza sentinel surveillance was established in 2008 and recently 16 severe acute respiratory illness (SARI) and 4 influenza-like illness (ILI) sites are engaged in the activities. The use of real-time PCR assays now allows for the simultaneous detection of a broad range of pathogens from a single respiratory specimen.

There are various advantages to studying SARS-CoV-2 co-infections with other respiratory viruses. Firstly, in high-risk COVID-19 patients (older adults, children, and COPD patients), co-infection can exacerbate the condition and raise the chance of death [3, 16] A severe inflammatory process that results in lung damage, an extended hospital stays, a change in the course and length of treatment, and a higher death rate are just some of the clinical implications associated with increased complications [11, 17, 18]. In addition, co-infections in outpatients may alter the way respiratory pathogens spread in community settings, putting vulnerable groups

like children and the elderly at risk and enabling viruses to infect families and community groups [11]. Therefore, early detection of co-infection may help to initiate proper management to avoid unnecessary complications and to reduce the transmission.

Despite the fact that a large number of studies have been undertaken, most of them were carried out outside of African countries and mostly focused on co-detection with the influenza virus A/B [4–10]. Therefore, to the best of our knowledge, this is the first descriptive analysis of respiratory viral co-detection from the analysis of a national SARI/ILI sentinel surveillance data in Ethiopia.

## Methods and materials

### Surveillance sites and study population

Starting from the establishment of the Ethiopian National Influenza Reference Laboratory in Ethiopia Public Health Institute (EPHI), Ethiopia has launched influenza sentinel surveillance since 2008, and currently, fourteen SARI and four ILI sites are engaged in SARS-CoV-2, influenza, and respiratory syncytial virus (RSV) surveillance system.

The selection of SARI and ILI sentinel surveillance sites in Ethiopia was intended to represent different geographical locations in 9 regional states and 2 city administrations throughout the country. It is carried out by the Federal Ministry of Health/EPHI through a system which has support from CDC and WHO and extends from EPHI down to the sentinel site [19]. The four ILI sentinel sites, Shiromeda Health Center, Kolfe Health Center, Akaki Health Center, and Dil-fire Health Center are located in Addis Ababa and sixteen SARI sentinel sites (hospitals that monitors admitted patients with SARI cases) are located across all regions of Ethiopia.

### Case inclusion criteria

ILI and SARI cases were screened according to the WHO case definition of ILI and SARI. Thus, any outpatient with an acute respiratory illness with a temperature ≥38˚C and cough, within 10 days of symptoms onset, was eligible for influenza-like illness (ILI) enrolment. SARI is considered as an acute respiratory infection with a measured or history of fever ≥38˚C and cough, with the onset of symptoms within the past 10 days that required hospitalization [20–22].

Finally, we conducted an analysis of data collected from outpatients with ILI and inpatients with SARI from January 2021 through June 2022.

### Specimen collection and transportation

Each sentinel site has a protocol to collect throat or nasopharyngeal swab specimens as part of the sentinel surveillance system and each focal person at the sentinel site are well trained on specimen collection, management, and transportation. The sentinel site focal persons obtain verbal consent from each patient prior to the collection of samples. Throat/nasopharyngeal swab samples were systematically collected from outpatients of all ages that fulfilled the case definition for ILI per week at sentinel surveillance sites. Similarly, throat/nasopharyngeal swab samples were also collected from all consenting patients who fulfilled the SARI case definition and were admitted to designated SARI sentinel surveillance sites (hospitals). The specimens were collected within 10 days after the first onset of symptoms for ILI and SARI. The collected samples were placed in viral transport media (VTM) and stored at 4˚C until transported by trained postal service officers to the National Influenza Central Laboratory (NICL) at Ethiopian Public Health Institute (EPHI) [20, 22].

Shipment of the specimens in viral transport media to the NICL at EPHI was conducted within 72 hours of collection using a cold chain system. Viral RNA from the swabs were extracted and subjected to real-time PCR amplification with parameters set for influenza testing, according to the Center for Disease Control and Prevention (CDC) protocol using reagents obtained from International Reagent Resource (IRR).

## Molecular detection of SARS-CoV-2 and other viral respiratory viruses

Viral RNA was extracted from throat/ nasopharyngeal swab samples using MagaBio plus Virus RNA Purification Kit II by MGISP-NE32 automated extractor which enables extracting 32 samples at a time within less than 10 minutes. The extracted RNA was detected by the real-time reverse transcriptase polymerase chain reaction (rRT_PCR) kit obtained via IRR from CDC Atlanta, for qualitative detection and differentiation of influenza viruses and Respiratory Syncytial Virus. The first step of the assay detects virus type as influenza A/influenza B/ RSV and the second step differentiates between influenza virus subtypes. The assay has positive control and primers and probes against the target genes for each virus. For SARS-CoV-2 BGI detection kit from China was used with positive control. ABI 7500 Fast PCR machine was used for detection [22].

## Data analysis

The collected data were encoded in Microsoft Excel combining the results of SARS-CoV-2 and other respiratory virus assays and data obtained from the patient record was exported to Stata version 16 software for analysis. Prior to statistical analyses, all personal identification data were eliminated. Descriptive statistics was carried out to determine the frequency and proportions of the categorical variables and the mean/median and standard deviations/inter-quartile ranges (IQR) of the continuous variables. Factors associated with SARS-CoV-2 co-detection were determined first by bivariable logistic regression, which was followed by multivariable logistic regression analysis. Those variables with a P-value <0.25 in the bivariable analysis were used as candidate variables for the multivariable analysis. The outcome variable used for the regression analysis was SARS-CoV-2 co-detection (yes or no) and the explanatory variables include the age of the patient, sex, specimen type, and region. Age of the patients was grouped based on the WHO age classification for influenza patients [19]. The findings of the multivariable analysis were expressed by a 95% confidence interval (CI) and adjusted odds ratio (AOR). The level of significance was set at 5%.

This analysis was done using data collected as part of routine surveillance activities, and as such, ethical approval was deemed not necessary by the Ethiopian Public Health Institute's Scientific and Ethical Review Office (SERO). Besides, data used in this article were collected according to the national respiratory viral diseases surveillance protocol which also include the consent of the participant/guardian before the collection; however, participants did not provide written informed consent for the use of their surveillance records in this study. Thus, data were de-identified and accessed for research purpose on September 30, 2022 and only code numbers were utilized the entire time.

## Results

### Characteristics of study participants

Data from a total of 4,989 ILI/SARI patients over 18 months period from January 01, 2021, to June 30, 2022 was analyzed (S1 Fig). Three thousand three hundred one (66.2%) and 1,688 (33.8%) were patients with SARI and ILI, respectively. A slight preponderance of them were males (52.6%) and residents of Addis Ababa (52%) (Table 1). The median age of the study population was 14 years (IQR: 1–35 years).

**Table 1. Demographic characteristics of ILI/SARI cases visited sentinel sites from January 2021 to June 2022, Ethiopia.**

| Variables | Category | Frequency | Percent |
|---|---|---|---|
| Age (in years) | 0-<2 | 1,271 | 25.5 |
| | 2-<5 | 700 | 14.0 |
| | 5-<15 | 545 | 10.9 |
| | 15-<50 | 1,714 | 34.4 |
| | 50-<65 | 448 | 9.0 |
| | ≥65 | 311 | 6.2 |
| Sex | Male | 2,626 | 52.6 |
| | Female | 2,363 | 47.4 |
| Region | Addis Ababa | 2,595 | 52.0 |
| | Dire Dawa | 466 | 9.3 |
| | Afar | 367 | 7.4 |
| | Sidama | 363 | 7.3 |
| | SNNPR | 348 | 7.0 |
| | Oromia | 257 | 5.2 |
| | Amhara | 203 | 4.1 |
| | Beneshangul Gumuz | 194 | 3.9 |
| | Somali | 167 | 3.4 |
| | Gambella | 29 | 0.6 |
| Type of disease | SARI | 3,301 | 66.2 |
| | ILI | 1,688 | 33.8 |

### SARS-CoV-2 co-detection with other respiratory viruses

Among the total samples tested (4,989), 459 (9.2%, 95% CI: 8.4–10.0) were positive for SARS-CoV-2 only and 64 (1.3%, 95% CI:1.0–1.6) had co-infection of SARS-CoV-2 and other respiratory viruses. From the total co-infections, 35(54.7%), 21(32.8%), and 8(12.5%) were with influenza virus, human respiratory syncytial virus (RSV), and both influenza and RSV, respectively (S1 Table). The distribution of co-detections with age showed that more than half of them 35(54.7%) were within the 15-<50 age category followed by children with age less than two years old 10 (15.6%). Regarding sex and type of cases, 38(59.4%) were females and 51 (79.7%) were from ILI patients (Table 2).

### Factors associated with SARS-CoV-2 and other respiratory viruses' co-detection

Three variables were included in the final multivariable analysis (S1 Dataset). The Odds of viral co-detection among SARI cases from other Ethiopian regions as compared to that of both SARI and ILI cases from Addis Ababa is 0.16. Regarding the odds of co-detection across different age groups, the odds of SARS-CoV-2 co-detection was about two times higher among the age group 20 to 39 years old as compared to the age group of 0 to 19 years old. The sex of participants was not associated with the SARS-CoV-2 co-detections when it was adjusted for the other factors (Table 3).

## Discussion

This study describes the magnitude of co-detection of SARS-CoV-2 with other respiratory viral infections and other associated factors among 4, 989 cases screened during 18 months at both ILI and SARI sentinel surveillance sites in Ethiopia. The overall proportion of co-

**Table 2. Characteristics of patients with SARS-CoV-2 and other respiratory viruses' co-detection.**

| Variables | Frequency of co-detection | Percentage (%) |
|---|---|---|
| Age | | |
| 0- <2 | 10 | 15.6 |
| 2 - <5 | 8 | 12.5 |
| 5 - <15 | 6 | 9.4 |
| 15 - <50 | 35 | 54.7 |
| 50 - <65 | 2 | 3.1 |
| ≥ 65 | 3 | 4.7 |
| Sex | | |
| Male | 26 | 40.6 |
| Female | 38 | 59.4 |
| Region | | |
| Addis Ababa | 56 | 87.5 |
| Other regions | 8 | 12.5 |
| Disease type | | |
| ILI | 51 | 79.7 |
| SARI | 13 | 20.3 |

detection found in this study was 1.3% with 95%CI (0.1–1.6). This is in line with estimates from Chicago (1.6%) [23], Singapore (1.4%) [10], Barcelona (0.6%) [24], Northern California, United States (0.9%) [7] and China (0.4%) [25] but higher than that was reported from India (0.04%) [3]. We found that more than half (54.7%) of the SARS-CoV-2 co-detection in our study was occurring with the Influenza virus. The result from a systematic review and meta-analysis of 26 studies done by Dadashi and his colleagues reported the prevalence of SARS-CoV-2 co-infection with Influenza virus was 0.8% [16]. On the other hand, the finding of this study was lower than the one from Turkey (2.0%) [26], Indonesia [27] and New York (2.0%) [28]. The variation in the magnitude might be attributable to sample size, seasonality, vaccination coverage and geographic variability in respiratory pathogens.

Studies have reported that SARS-CoV-2 co-infections with other respiratory virus might be responsible for different types of clinical outcomes, which include either improve, aggravate,

**Table 3. Multivariable analysis of factors associated with SARS-CoV-2 co-infection with other respiratory viruses among ILI/SARI cases visited sentinel sites from January 2021 to June 2022, Ethiopia.**

| Variables | SARS-CoV-2 co-detection | | Crude OR (95% CI) | AOR (95% CI) | P—Value |
|---|---|---|---|---|---|
| | No | Yes | | | |
| Age (in years) | | | | | |
| 0–19 | 2,729 | 29 | 1 | 1 | |
| 20–39 | 1,115 | 25 | 2.11(1.23–3.62) | 1.98(1.15–3.42) | 0.014* |
| ≥40 | 1,081 | 10 | 0.87(0.42–1.79) | 1.27(0.61–2.64) | 0.528 |
| Sex | | | | | |
| Male | 2,600 | 26 | 1 | | |
| Female | 2,325 | 38 | 1.63(0.99–2.70) | 1.39(0.84–2.31) | 0.200 |
| Region | | | | | |
| Addis Ababa | 2,539 | 56 | 1 | | |
| Others | 2,386 | 8 | 0.15(0.07–0.32) | 0.16 (0.07–0.34) | < 0.001* |

* P<0.05 significantly associated variables

or no effect on clinical outcomes [29]. On the other hand, a study conducted by Pinky and his colleagues showed that SARS-CoV-2 infections are easily suppressed when initiated simultaneously or after infection with another respiratory virus. The finding suggests that each type of virus replication can be determined by the species and the growth rate of the virus [30]. Overall, recent evidence verify that respiratory viruses compete with one another [27].

In this study, we found that more than half of co-infected individuals were between the ages of 15 and 50 and females by gender. Patients between 20 and 39 were about twice as likely to have SARS-CoV-2 co-detection with other respiratory viral infections as compared to the age category of 0 to 19. This might be due to variability in susceptibility to SARS-CoV-2 infection among different age categories [31, 32]. A study conducted to estimate susceptibility to SARS-CoV-2 infection among different age categories by using age-specific case data from 32 settings and data from six studies showed that the susceptibility to SARS-CoV-2 infection in individuals under 20 years of age is approximately half that of adults aged over 20 years. Besides, the median age of co-detected patients was 20.5 years, younger than those only infected with SARS-CoV-2. This finding agrees with the body of research notion showing that younger populations are more likely to get community-acquired viral co-infections [33]. Furthermore, the difference in co-infection rate between males and females is important to emphasize as women are the primary caregivers of sick children in most populations and have higher risks of exposure.

Our finding indicates the SARS-CoV-2 co-detection rate was significantly higher among cases from Addis Ababa city compared to cases from other regions of the country (P <0.001). In this study, all tests had the same probability of detecting co-infection, irrespective of the region they were collected in. Our data set suggests that the difference in the co-detection rate might be due to the time that the individual visited health facilities to seek treatment rather than their residence location. Because two-thirds of the cases from Addis Ababa city originated from ILI sentinel sites (health centers), while all cases for the other regions were from SARI sentinel sites (hospitals). In contrast to SARI cases, which may take a few extra days to be admitted to hospitals because they may need a referral from a lower health unit, ILI cases may attend health facilities as early as possible after the onset of the symptoms. According to a systematic review conducted by Mallett and his colleagues, the virus detection percentage varies depending on the period, with the highest proportion occurring from nasopharyngeal sampling between 0 and 4 days following the onset of symptoms (89%), and the lowest percentage occurring between 10 and 14 days later (54%) [34].

We accept that our study was subject to some limitations. Only co-detections of viral agents —SARS-CoV-2 with influenza and RSV viruses were examined. Secondly, we did not have access to medical records to evaluate clinical data on comorbidity and monitor patient outcomes.

However, our study has strengths since it presents information from 18 Ethiopian sentinel sites that are geographically dispersed. Additionally, the majority of studies published globally have focused on co-detections in hospitalized patients with a small sample size and limited to particular age groups (children or elderly age category), but in our study, we addressed both outpatients and inpatients without any restrictions on age with a sizeable sample in Ethiopia. Multiple respiratory virus co-detections may indicate the possibility of viral competition and inhibition. Additionally, it shows that testing a specimen for many viruses is necessary rather than focusing on a specific infection and excluding other probable culprits during therapy [35, 36]. To evaluate the impact of the SARS-CoV-2 and influenza co-infection on clinical outcomes, including other epidemiological data, additional longitudinal studies are in fact required.

## Conclusions

This study found a 1–3% co-detection rate of SARS-CoV-2 and other respiratory viral infections. When compared to SARI inpatients, it is considerably detected in ILI outpatients. Furthermore, we observed that younger age groups accounted for the majority of co-detections. Therefore, it's crucial to identify concurrent viral respiratory pathogens as soon as possible in order to decrease community transmission, as well as to enhance diagnosis, clinical care, and patient prognosis. To establish the pathogenic impact of viral and bacterial co-infection in patients with and without co-infection, more research is required.

## Supporting information

**S1 Table. Distribution of SARS-CoV-2 co-detection and mono-infections across SARI/ILI sentinel sites from January 01, 2021, to June 30, 2022, Ethiopia.**
(XLSX)

**S1 Fig. Monthly trends of proportions of co-detection among SARS-CoV-2 positive cases from January 01, 2021, to June 30, 2022, Ethiopia.**
(TIF)

**S1 Dataset. Data set of the study.**
(XLSX)

## Acknowledgments

We would like to express our gratitude to all sentinel surveillance sites and the personnel that operate there for their crucial contribution to the ILI/SARI surveillance. We are grateful to the World Health Organization (WHO) and the United States Center for Disease Control (CDC) for their support to the surveillance system to have lab supplies and reagents. We also appreciate Ohio-State University, ICAP at Columbia University and EPHI for the support provided during the manuscript write-up.

## Author Contributions

**Conceptualization:** Wolde Shure, Adamu Tayachew, Tsegaye Berkessa, Gizaw Teka, Mengistu Biru, Ayele Gebeyehu, Musse Tadesse, Bizuwork Haile, Beza Addis, Muluken Moges, Lehageru Gizachew, Eyasu Tigabu, Getnet Yimer, Nebiyu Dereje, Jemal Aliy, Ebba Abate, Mesfin Wossen, Aschalew Abayneh.

**Data curation:** Wolde Shure, Adamu Tayachew, Tsegaye Berkessa, Gizaw Teka, Mengistu Biru, Ayele Gebeyehu, Adane Woldeab, Musse Tadesse, Melaku Gonta, Admikew Agune, Aster Hailemariam, Leuel Lisanwork, Lehageru Gizachew, Eyasu Tigabu, Zelalem Mekuria, Getnet Yimer, Nebiyu Dereje, Jemal Aliy, Sileshi Lulseged, Zenebe Melaku, Ebba Abate, Wondwossen Gebreyes, Mesfin Wossen, Aschalew Abayneh.

**Formal analysis:** Wolde Shure, Tsegaye Berkessa, Ayele Gebeyehu, Adane Woldeab, Lehageru Gizachew, Nebiyu Dereje, Jemal Aliy, Mesfin Wossen.

**Investigation:** Wolde Shure, Adamu Tayachew, Tsegaye Berkessa, Gizaw Teka, Mengistu Biru, Ayele Gebeyehu, Adane Woldeab, Musse Tadesse, Melaku Gonta, Admikew Agune, Aster Hailemariam, Bizuwork Haile, Beza Addis, Leuel Lisanwork, Eyasu Tigabu, Jemal Aliy, Sileshi Lulseged, Zenebe Melaku, Wondwossen Gebreyes, Mesfin Wossen, Aschalew Abayneh.

**Methodology:** Wolde Shure, Adamu Tayachew, Tsegaye Berkessa, Gizaw Teka, Ayele Gebeyehu, Adane Woldeab, Musse Tadesse, Melaku Gonta, Admikew Agune, Aster Haile-mariam, Bizuwork Haile, Beza Addis, Muluken Moges, Leuel Lisanwork, Lehageru Giza-chew, Eyasu Tigabu, Nebiyu Dereje, Sileshi Lulseged, Zenebe Melaku, Ebba Abate, Wondwossen Gebreyes, Mesfin Wossen.

**Resources:** Gizaw Teka, Mengistu Biru, Leuel Lisanwork, Lehageru Gizachew, Zelalem Mekuria, Getnet Yimer, Jemal Aliy, Ebba Abate, Wondwossen Gebreyes, Mesfin Wossen, Aschalew Abayneh.

**Software:** Wolde Shure, Adamu Tayachew, Tsegaye Berkessa, Gizaw Teka.

**Supervision:** Wolde Shure, Adamu Tayachew, Tsegaye Berkessa, Gizaw Teka, Mengistu Biru, Ayele Gebeyehu, Admikew Agune, Aster Hailemariam, Muluken Moges, Lehageru Gizachew.

**Validation:** Wolde Shure, Adamu Tayachew, Tsegaye Berkessa, Gizaw Teka, Zelalem Mekuria, Nebiyu Dereje, Mesfin Wossen, Aschalew Abayneh.

**Visualization:** Wolde Shure, Adamu Tayachew, Tsegaye Berkessa, Gizaw Teka, Lehageru Gizachew, Ebba Abate.

**Writing – original draft:** Wolde Shure, Adamu Tayachew, Tsegaye Berkessa, Gizaw Teka, Mengistu Biru, Ayele Gebeyehu, Adane Woldeab, Melaku Gonta, Leuel Lisanwork, Leha-geru Gizachew, Eyasu Tigabu, Zelalem Mekuria, Getnet Yimer, Nebiyu Dereje, Jemal Aliy, Sileshi Lulseged, Zenebe Melaku, Ebba Abate, Wondwossen Gebreyes, Mesfin Wossen, Aschalew Abayneh.

**Writing – review & editing:** Wolde Shure, Adamu Tayachew, Tsegaye Berkessa, Gizaw Teka, Mengistu Biru, Ayele Gebeyehu, Adane Woldeab, Musse Tadesse, Melaku Gonta, Admikew Agune, Aster Hailemariam, Bizuwork Haile, Beza Addis, Muluken Moges, Leuel Lisan-work, Lehageru Gizachew, Eyasu Tigabu, Zelalem Mekuria, Getnet Yimer, Nebiyu Dereje, Jemal Aliy, Sileshi Lulseged, Zenebe Melaku, Ebba Abate, Wondwossen Gebreyes, Mesfin Wossen, Aschalew Abayneh.

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
