## [Decision Letter · Decision Letter 0]

6 Dec 2023

PGPH-D-23-01866

SARS-CoV-2 Co-detection with Influenza and Human Respiratory Syncytial Virus in Ethiopia: Findings from the severe acute respiratory illness (SARI) and influenza-like illness (ILI) sentinel surveillance, January 01, 2021, to June 30, 2022

Dear Dr. Berkessa,

Thank you for submitting your manuscript to PLOS Global Public Health. After careful consideration, we feel that it has merit but does not fully meet PLOS Global Public Health’s publication criteria as it currently stands. Therefore, we invite you to submit a revised version of the manuscript that addresses the points raised during the review process.

We look forward to receiving your revised manuscript.

Kind regards,

Martha I. Nelson

Academic Editor

Journal Requirements:

1. We noticed you have some minor occurrence of overlapping text with the following previous publication(s), which needs to be addressed:

https://www.frontiersin.org/articles/10.3389/fmed.2021.681469/full

https://bmjopen.bmj.com/content/11/11/e053768.full

https://bmcinfectdis.biomedcentral.com/articles/10.1186/s12879-020-4827-0

In your revision ensure you cite all your sources (including your own works), and quote or rephrase any duplicated text outside the methods section. Further consideration is dependent on these concerns being addressed.

Additional Editor Comments (if provided):

Reviewers' comments:

Reviewer's Responses to Questions

**Comments to the Author**

1. Does this manuscript meet PLOS Global Public Health’s publication criteria? Is the manuscript technically sound, and do the data support the conclusions? The manuscript must describe methodologically and ethically rigorous research with conclusions that are appropriately drawn based on the data presented.

Reviewer #1: Yes

Reviewer #2: Yes

2. Has the statistical analysis been performed appropriately and rigorously?

Reviewer #1: N/A

Reviewer #2: Yes

3. Have the authors made all data underlying the findings in their manuscript fully available (please refer to the Data Availability Statement at the start of the manuscript PDF file)?

Reviewer #1: No

Reviewer #2: No

4. Is the manuscript presented in an intelligible fashion and written in standard English?

Reviewer #1: Yes

Reviewer #2: Yes

5. Review Comments to the Author

Reviewer #1: The study offers valuable insights into the co-detection of SARS-CoV-2 with other respiratory viruses. The exploration of age-related patterns and the call for additional research on pathogenic impacts add depth to the findings. Here are comments that may help improve the paper:

Comments:

•Ensure consistency in terminology; for instance, the manuscript refers to “SARS-COV-2” and “SARS-CoV-2”; choose one consistent term throughout.

•The abstract is well structured, although some refinements in language precision and consistency would enhance its clarity.

•In line 100-101 there is a typographical error “real-time PCR”.

•In line 87-89, please provide specific data or references, such as the rate of co-infections.

•Consider briefly acknowledging key findings or trends from studies conducted in other parts of the world.

•Mention the significance of the study within the Ethiopian context. How will the findings contribute to addressing the local challenges, especially in comparison to studies conducted elsewhere?

•Consider providing a brief rationale for choosing specific variable in the regression analysis, which would enhance the understanding.

•The shift in age categories between Table 1-2 and Table 3 raises a notable point. While Table 2 provides a detailed breakdown of age groups, the transition in Table 3 to broader categories may impact the precision of age-related findings. A transparent explanation of the rationale behind the change in categorization would strengthen the interpretation of age-related associations in the study.

•In line 192, most likely this is likely a typographical error in the sentence “under two age category10”.

•In the multivariable analysis, the number of “yes” cases are relatively low compared to the “no” cases, raising considerations about the stability and precision of the estimated parameters. With this substantial imbalance in the number of cases between groups, there is a risk of limited statistical power, which may affect the reliability of the results and increase the potential for type II errors. The author might explore alternative methods or approaches to mitigate the impact of imbalanced groups in the analysis.

•Add a supplementary table to support the reported overall co-detection proportion (1.3%)

•In line 207, there is an extra comma 0.9%

•In line 208, what was the percentage for India?

•For a more comprehensive context, it would be beneficial for the authors to include the total number of inpatients and outpatients in the study.

Reviewer #2: Could you show any plots of SARI/ILI/SARS-CoV-2/influenza/RSV circulation over time? Was there a time period when the epidemics overlapped and coinfection was more common? What was the seasonality of these infections over this 18-month time period? Did SARS-CoV-2 alter the expected seasonal timing of influenza or RSV? Was there a larger flu or RSV epidemic than usual, like there was in Western countries where lockdowns meant no flu or RSV activity for much of 2020-2021? Did these viruses rise and fall at the same time across all 14 sites?

It would be nice to have a map of the location of the study sites, shaded by population density or annotated with major cities.

The methods say that the assay distinguishes influenza A and influenza B. Was there enough power to compare flu A and B?

6. PLOS authors have the option to publish the peer review history of their article (what does this mean?). If published, this will include your full peer review and any attached files.

**Do you want your identity to be public for this peer review?** For information about this choice, including consent withdrawal, please see our Privacy Policy.

Reviewer #1: No

Reviewer #2: No

---

## [Decision Letter · Decision Letter 1]

19 Mar 2024

SARS-CoV-2 Co-detection with Influenza and Human Respiratory Syncytial Virus in Ethiopia: Findings from the severe acute respiratory illness (SARI) and influenza-like illness (ILI) sentinel surveillance, January 01, 2021, to June 30, 2022

PGPH-D-23-01866R1

Dear Mr. Berkessa,

We are pleased to inform you that your manuscript 'SARS-CoV-2 Co-detection with Influenza and Human Respiratory Syncytial Virus in Ethiopia: Findings from the severe acute respiratory illness (SARI) and influenza-like illness (ILI) sentinel surveillance, January 01, 2021, to June 30, 2022' has been provisionally accepted for publication in PLOS Global Public Health.

Best regards,

Martha I. Nelson

Academic Editor

Your paper has been accepted for publication at PLOS Global Public Health. In the future, if you could please format your response to reviewer comments a little differently, it would speed the second review process and save everyone time. By this I mean include quoted text and line numbers in each response, so the reviewer can easily cross reference the response back to the revised manuscript. For example:

Reviewer Comment: Please do X.

Author Response: We have done X in lines 100-121 of the revised manuscript, see quoted text below.

["New text from lines 100-121."]

Otherwise it takes reviewers a lot of time to sort through all the responses and delays the review process.

Reviewer Comments (if any, and for reference):

Reviewer's Responses to Questions

**Comments to the Author**

1. If the authors have adequately addressed your comments raised in a previous round of review and you feel that this manuscript is now acceptable for publication, you may indicate that here to bypass the “Comments to the Author” section, enter your conflict of interest statement in the “Confidential to Editor” section, and submit your "Accept" recommendation.

Reviewer #1: All comments have been addressed

Reviewer #2: All comments have been addressed

2. Does this manuscript meet PLOS Global Public Health’s publication criteria? Is the manuscript technically sound, and do the data support the conclusions? The manuscript must describe methodologically and ethically rigorous research with conclusions that are appropriately drawn based on the data presented.

Reviewer #1: Yes

Reviewer #2: Yes

3. Has the statistical analysis been performed appropriately and rigorously?

Reviewer #1: Yes

Reviewer #2: Yes

4. Have the authors made all data underlying the findings in their manuscript fully available (please refer to the Data Availability Statement at the start of the manuscript PDF file)?

Reviewer #1: No

Reviewer #2: Yes

5. Is the manuscript presented in an intelligible fashion and written in standard English?

Reviewer #1: Yes

Reviewer #2: Yes

6. Review Comments to the Author

Reviewer #1: The manuscript is well-organized and provides a comprehensive analysis of SARS-CoV-2 co-detection with other respiratory viruses in Ethiopia. The authors have addressed all previous comments and suggestions. The introduction sets the context, highlighting the research gap. The methods are detailed, outlining surveillance sites, criteria, and laboratory procedures. Results are clearly presented, covering prevalence rates and associated factors. The discussion interprets findings and suggests future directions. Overall, it's a robust study contributing valuable insights to the field.

Reviewer #2: (No Response)

7. PLOS authors have the option to publish the peer review history of their article (what does this mean?). If published, this will include your full peer review and any attached files.

**Do you want your identity to be public for this peer review?** For information about this choice, including consent withdrawal, please see our Privacy Policy.

Reviewer #1: No

Reviewer #2: No
